# A Step towards Efficient Waste Segmentation for Automated Waste Recycling in Cluttered Background

## Abstract

Rapid expansion of urban areas and population growth is causing an immense increase in waste production, which demands the need for efficient and automated waste management. In this scenario, automated waste recycling that utilizes deep learning methods to separate the recyclable waste objects may emerge as a savior to humanity. Recent deep learning approaches for automated waste recycling provide promising waste segmentation performance in cluttered scenarios. However, these methods rely on large backbone networks that are inefficient for automated waste recycling systems. To this end, we propose an efficient waste segmentation network, where the spatial context module enhances localized structural dependencies in the spatial domain, and the spectral context module subsequently captures global contextual relationships in the frequency domain. This cascaded design allows the network to progressively leverage both local and global representations across complementary domains to highlight the semantic information necessary for effective segmentation of waste objects in cluttered scenes. Furthermore, our auxiliary feature enhancement focuses on structural information to enhance the target objects' boundaries and blob amplification for better segmentation. Extensive experimentation on two challenging waste segmentation datasets including *ZeroWaste-f* and *SpectralWaste* reveals the merits of the proposed method.

## 1 Introduction

Recent studies state that annual waste generation can reach to an estimate of three billion tons by 2050 Kaza et al. (2018). This drastic increase in solid waste production will have destructive effects on the ecosystem and requires careful considerations for waste disposal and recycling Subedi et al. (2025). The current advancement in artificial intelligence techniques encourages humans to perform automated separation of recyclable waste items from solid waste. This enables the recycling plants to quickly process the waste material without exposing the manpower to pointed and unhygienic waste objects. In this scenario, several waste classification and sorting frameworks Mao et al. (2021); Feng et al. (2022) have been introduced that utilize convolution neural networks for waste classification. These methods are marginally practical and require more sophisticated approaches that can detect the solid waste and separate it. To this extent, waste detection methods are proposed that can effectively detect the waste objects in a scene. For example, Xia et al. Xia et al. (2024) introduced a lightweight YOLO model to detect the garbage items in densely accumulated scenarios. Sun et al. Sun et al. (2025) proposed a framework based on YOLOv10n for household garbage detection. Such works provide reasonable performance for detection in moderately complex scenarios, however, their performance degrades considerably in more challenging scenes such as the presence of translucent and varying-sized deformable waste objects in a cluttered environment.

Considering that the need for effective and efficient waste detection and segmentation frameworks is crucial, Bashkirova et al. Bashkirova et al. (2022) introduced ZeroWaste dataset that provides the opportunity to the research community to improve the performance of the waste detection and segmentation networks. The authors demonstrated that existing popular segmentation methods Chen et al. (2018); Liu et al. (2021); Ouali et al. (2020) strive to accurately segment various waste items in cluttered scenes. Later on, Casao et al. Casao et al. (2024) presented the SpectralWaste dataset that further pose significant challenges by including thin and elongated waste items such as video tape and filament. To handle these challenges, Ali et al. Ali et al. (2024) proposed FANet that utilizes adaptive feature enhancement layer to highlight the

object boundaries and improve segmentation results. Subsequently, COSNet Ali et al. (2025) is introduced that reduces the computational cost while improving waste segmentation performance. Although effective, these methods are inspired from spatial enhancement convolution filters that only capture small neighborhood context and become computationally more expensive when the filter size is increased to capture global relationships between features. Additionally, these methods need further optimization for practical use in automated waste recycling. To this end, we propose an efficient waste segmentation network (EWSegNet), where spatial features are first refined to capture local dependencies, followed by spectral features that model global frequency relationships. The sequential design enables the network to leverage complementary context across domains efficiently. Moreover, by incorporating structural information and blob amplification, our auxiliary feature enhancement strengthens object boundaries. It highlights key regions, resulting in better segmentation performance without degrading the efficiency of the model. We summarize the key contributions of our work as follows:

- We propose a novel end-to-end efficient waste segmentation network, called EWSegNet, which improves the model's computational efficiency without compromising the segmentation performance.

- We propose the frequency context module (FCM) that enables the model to learn data-dependent kernel and capture global context for better segmentation results.

- We introduce the spatial context module (SCM) that performs the feature excitation and weighting to complement the FCM for rich feature extraction.

- To emphasize the boundaries of the waste items and blob regions, we design auxiliary feature enhancement module (AFEM) that uses difference of Gaussian filtering and pooled attention thereby improving waste segmentation performance.

- We perform extensive experimentation on two challenging waste segmentation datasets and demonstrate the effectiveness of the proposed method.

## 2 RELATED WORK

This section briefly reviews the existing work related to waste identification and segmentation methods. In addition, we discuss image and boundary enhancement techniques in spatial and frequency domains.

**Waste Detection and Segmentation Methods:** With the advancement of deep learning approaches, automated waste recognition and detection became more demanding from industrial perspective. For instance, Mao et al. Mao et al. (2021) utilized DenseNet Huang et al. (2017) for waste classification necessary to perform recycling tasks, Feng et al. Feng et al. (2022) introduced EfficientNet Koonce (2021) based method for automated waste classification and sorting, Meng et al. Meng et al. (2022) demonstrated the use of MobileNet Howard et al. (2017) for waste detection, Tian et al. Tian et al. (2024) proposed efficient garbage classification method, and Xia et al. Xia et al. (2024) introduced a light-weight YOLO model for garbage detection that utilizes MobileVit-v3 Wadekar & Chaurasia (2022) for feature extraction. Despite the fact that these methods perform favorably for waste classification and detection, their performance is degraded in cluttered scenes. Moreover, recent waste detection and segmentation datasets Bashkirova et al. (2022); Casao et al. (2024) pose significant challenges to existing methods due to scale variations, presence of translucent and deformable objects, and cluttered background. Bashkirova et al. Bashkirova et al. (2022) demonstrated that existing detection and segmentation methods He et al. (2017); Li et al. (2019); Chen et al. (2018); Liu et al. (2021); Ouali et al. (2020) struggle to detect and segment translucent and deformable waste objects in cluttered scenes. Subsequently, Ali et al. Ali et al. (2024) proposed FANet that utilizes the feature refinement module to enhance the features for better segmentation of translucent objects. More recently, COSNet Ali et al. (2025) is introduced that utilizes sharpening module to emphasize the boundary details of the waste objects on cluttered scenes. Although these methods provide promising results, they have relatively high computational cost which makes them less appropriate for automated waste recycling. In contrast, we propose a waste segmentation network that is more efficient and operates in the spectral domain to enhance the feature maps thereby more desirable to be used in waste recycling systems.

**Image Enhancement and Sharpening:** Several image and boundary enhancement procedures exist in the literature that utilize the sharpening filters to emphasize edges and improve the contrast of the image. Most of the techniques operate in spatial domain and utilize predefined filters or kernels to sharpen the

Figure 1: Overall framework of the proposed efficient waste segmentation network (EWSegNet) is illustrated here. The encoder consists of four stages that provide multiscale feature representations (i.e., $F_1, F_2, F_3, F_4$). Each stage contains $N_i$ number of EWFE layers (where $i \in [1, 2, 3, 4]$). Before each stage, a convolution layer is used to downsample the feature maps. Feature representations of stage three are fed to AFEM to emphasize boundaries and blob regions and obtain feature maps $F_5$. Finally, these multiscale features are fed to decoder to obtain segmentation map.

edges and highlight the fine details Gonzalez & Woods (2018a). For example, *Laplacian* kernel, unsharp masking and high-boost filtering are some of the techniques that operate in spatial domain to sharpen the boundaries and enhance the image. However, utilizing these spatial domain kernels for convolution in deep neural networks is tricky. Additionally, spatial domain filtering is suitable for small-sized kernels, as computational cost and parameters increase significantly with increasing kernel size. Alternatively, image enhancement in frequency domain is fairly easier as convolution of a function $f(x)$ with a kernel $h(x)$ in spatial domain is equivalent to multiplication in frequency domain Gonzalez & Woods (2018b) and is expressed as:

$$(f \star h)(x) \Leftrightarrow (H \cdot F)(\mu) \tag{1}$$

where $H$ and $F$ denote the Fourier transform of the corresponding functions. In addition, frequency domain provides more flexibility to design any type of kernel such as a high-pass or low-pass kernel. For instance, a high-pass kernel can be designed by using difference of Gaussian functions as follows.

$$H(u) = Ae^{\frac{-u^2}{2\sigma_1^2}} - Be^{\frac{-u^2}{2\sigma_2^2}} \tag{2}$$

where $A \geq B$ and $\sigma_1 > \sigma_2$. Inspired by frequency domain filtration, we design our Frequency Context module (FCM) and Auxiliary Feature Enhancement module (AFEM) which are discussed in Sec. 3.2.2 and 3.3 respectively.

## 3 METHOD

This section describes in detail the overall architecture of the proposed efficient waste segmentation network (EWSegNet).

### 3.1 OVERALL ARCHITECTURE

The general architecture of the proposed waste segmentation framework (EWSegNet) is shown in Figure 1. The framework consists of an encoder, an auxiliary feature enhancement module (AFEM), and a segmentation decoder. The EWSegNet encoder is responsible for extracting multiscale feature maps by utilizing the efficient waste feature extraction (EWFE) layers. The encoder is composed of four stages; each stage contains $N_i$ number of EWFE layers (where $i \in [1, 2, 3, 4]$). Before each stage, a convolution layer is utilized to decrease the spatial resolution of feature maps while increasing the number of channels. The first convolution layer is represented as a stem layer that is responsible for projecting the image to feature space and reducing the spatial resolution. Unlike other segmentation encoders, EWSegNet takes the features of

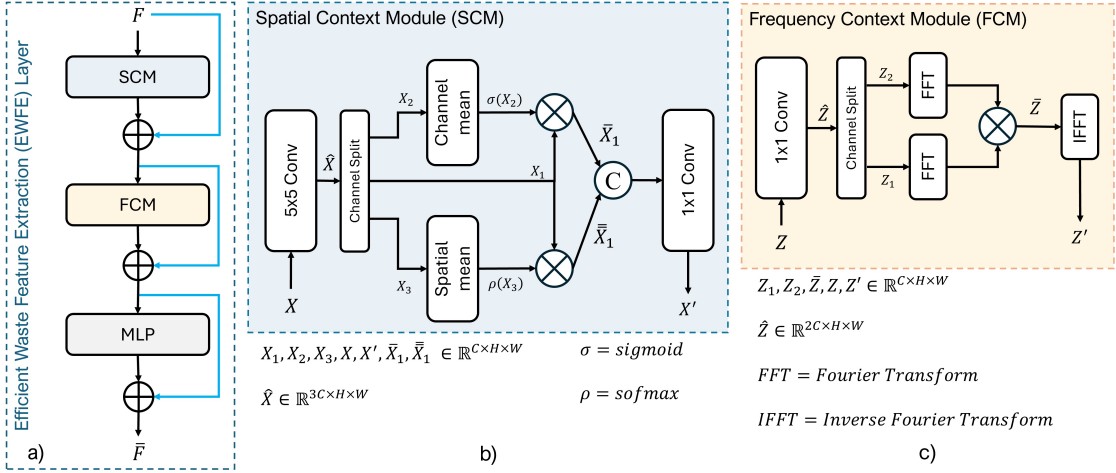

Figure 2: Efficient waste feature extraction (EWFE) layer is shown in Fig. a). Fig b) represents the spatial context module (SCM) that is used for feature excitation and weighting in spatial domain. In c), frequency context module (FCM) is illustrated that captures global contextual relationship between pixels in frequency domain.

the third stage and performs feature enhancement by means of an auxiliary feature enhancement module. Afterwards, these enhanced features are added back to the feature maps of the third stage and then fed to the fourth stage of the encoder. This operation encourages the network to focus on the boundary information necessary to segment the translucent waste objects in cluttered scenes and is described in detail in Sec. 3.3. The extracted multiscale feature maps of four stages and the enhanced feature maps from the AFEM are then fed to the segmentation decoder which processes these multiscale feature representations and provides the segmentation mask.

## 3.2 EFFICIENT WASTE FEATURE EXTRACTION LAYER

The basic building block of EWSegNet encoder is efficient waste feature extraction (EWFE) layer as shown in Fig. 2. The figure illustrates that EWFE consists of three blocks, including spatial context module (SCM), frequency context module (FCM), and a multilayer perceptron (MLP). The SCM and FCM blocks are responsible for efficiently capturing the semantic information in spatial and spectral domains. We discuss these blocks in more detail below.

### 3.2.1 SPATIAL CONTEXT MODULE

The main objective of the spatial context module (SCM) is to capture the contextual relationship of pixels within a local neighborhood. As illustrated in Fig. 2, it first utilizes the $5 \times 5$ group-wise convolution to project the input $X \in \mathbb{R}^{C \times H \times W}$ to $\hat{X}$ of shape $3C \times H \times W$. Then, it splits the features maps $\bar{X}$ in the channel dimension to obtain features $X_1, X_2$ and $X_3$ of shape $C \times H \times W$. The feature maps $X_2$ and $X_3$ are used to highlight the significant features in $X_1$. To achieve this, the mean is taken along the channel dimension of the features $X_2$ and *sigmoid* function ($\sigma$) is applied to obtain the feature weights which are then multiplied element-wise to get weighted features $\bar{X}_1$. Similarly, spatial mean is taken for the features $X_3$ and *softmax* function ($\rho$) is applied along the channel dimension. This vector is then multiplied with the features $X_1$ to highlight the important channels and obtain feature maps $\bar{\bar{X}}_1$. Finally, feature maps $\bar{X}_1$ and $\bar{\bar{X}}_1$ are concatenated along channel dimension and passed to a $1 \times 1$ convolution to obtain the weighted features $X'$. Mathematically, the whole operation is expressed as below:

$$
\begin{aligned}
X_1, X_2, X_3 &= CSplit(Conv_{5 \times 5}(X)) \\
\bar{X}_1 &= X_1 \cdot \sigma(CMean(X_2)) \\
\bar{\bar{X}}_1 &= X_1 \cdot \rho(SMean(X_3)) \\
X' &= Conv_{1 \times 1}(concat(\bar{X}_1, \bar{\bar{X}}_1))
\end{aligned}
\tag{3}
$$

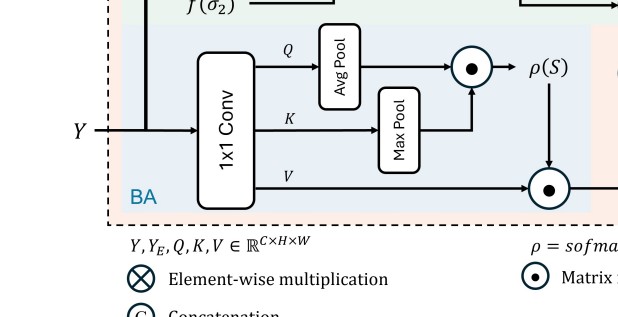

Figure 3: This figure demonstrates the auxiliary feature enhancement module (AFEM) that has dual functions: boundaries enhancement (BE) and blob amplification (BA). BE emphasizes the fine details by using difference of Gaussian filtration while BA uses pooled attention to focus on semantic regions.

where $CSplit$ refers to the channel-wise split, $SMean$ is spatial mean, $CMean$ is the mean along channel dimension, $\sigma$ represents sigmoid function, and $\rho$ denotes softmax function, respectively.

### 3.2.2 FREQUENCY CONTEXT MODULE

The aim of the frequency context module (FCM) is to capture the global relationship between the pixels which is illustrated in Fig. 2. To accomplish this efficiently, the input feature maps $Z \in \mathbb{R}^{C \times H \times W}$ are fed to $1 \times 1$ convolution to obtain feature maps $Z_1$ and $Z_2$ of shape $C \times H \times W$ respectively. Afterwards, Fourier transform is applied on the feature maps $Z_1$ and $Z_2$ to project the features in frequency domain. Then, the features in frequency domain are multiplied together to signify the important features and obtain feature maps $\bar{Z}$. As multiplication in frequency domain is equivalent to the convolution in spatial domain, therefore, this operation encourages the model to learn to focus on the significant information necessary for the required task. Eventually, the feature representations $\bar{Z}$ are projected back to spatial domain by taking the inverse Fourier transform and obtain rich feature representations $Z'$.

### 3.3 AUXILIARY FEATURE ENHANCEMENT MODULE

Auxiliary feature enhancement module (AFEM) is the critical component of EWSegNet as its objective is to enhance the boundary information and to emphasize the structural information and the blob regions. AFEM has two main components including boundary emphasis (BE) and blob amplification (BA) as demonstrated in Fig. 3. AFEM takes the features of third stage as input and emphasize the boundary information by utilizing difference of Gaussian filter in frequency domain. To accomplish this, input $Y$ is first transformed to frequency domain by means of Fourier transform and multiplied with two Gaussian functions having sigma values of $\sigma_1$ and $\sigma_2$ to obtain features $Y_{f1}$ and $Y_{f2}$ respectively. These feature maps are then projected back to spatial domain by using inverse Fourier transform and get features $Y_{i1}$ and $Y_{i2}$ respectively. Afterwards, difference is taken between them to get the high frequency information $H_f$ needed to focus on the boundary information. Additionally, spatial mean is taken on the input features $Y$ to get the channel weights $W_c$. These channel weights $W_c$ are multiplied with the boundary features $H_f$ to boost the relevant channels and obtain emphasized boundary features $B_e$. Finally, features $Y_{i2}$ are added to the extracted boundary information $B_e$ to obtain boundary emphasized feature representations $Y_B$.

To amplify the blob regions, first, the input $Y$ is passed to the $1 \times 1$ convolution layer to obtain feature maps $Q, K$ and $V$. Then, feature maps $Q$ are passed to average pooling layer to reduce the spatial dimension and

Table 1: Performance comparison of EWSegNet with state-of-the-art waste segmentation methods in terms of mIoU (%) and pixel accuracy (%) metrics on the test split of ZeroWaste-f dataset. Best results are highlighted in bold and second best are underlined.

| Method | Encoder Params (M) ↓ | Latency (msec) ↓ | mIoU (%)↑ | Pix. Acc. (%)↑ |
|---|---|---|---|---|
| DeepLabv3+ Chen et al. (2018) | - | - | 52.13 | 91.38 |
| FANet Ali et al. (2024) | 36.7 | 74.5 | 54.89 | 91.41 |
| FocalNet-B Yang et al. (2022) | 88.7 | - | 54.26 | 91.28 |
| COSNet Ali et al. (2025) | 27.3 | 73.6 | **56.67** | **91.91** |
| EWSegNet (Ours) | **23.3** | **64.8** | 56.44 | 91.75 |

Table 2: Performance comparison in terms of class-wise IoU (%) scores of EWSegNet with recent waste segmentation frameworks on ZeroWaste-f dataset. Best results are highlighted in bold text.

| Method | IoU (%) ↑ | | | | |
|---|---|---|---|---|---|
| | Background | Cardboard | Soft Plastic | Rigid Plastic | Metal |
| DeepLabv3+ | 91.02 | 54.47 | 63.18 | 24.82 | 27.14 |
| COSNet | 91.44 | 59.13 | **65.92** | **37.24** | 29.61 |
| EWSegNet (Ours) | **91.45** | **59.24** | 63.17 | 33.28 | **35.05** |

obtain averaged response in a $n \times n$ neighborhood. Similarly, feature maps $K$ are passed to a max-pooling layer to get the significant features within the $n \times n$ neighborhood. Afterwards, self-attention is computed between the pooled features and $V$ features to obtain the blob amplified features $Y_A$. Eventually, enhanced features $(Y_B, Y_A)$ are concatenated along the channel dimension and passed to $1 \times 1$ convolution layer to get the enhanced feature representations $Y_E$.

## 4 EXPERIMENTS

This section describes the datasets utilized in experimentation, evaluation metrics for performance comparison, and implementation details followed by quantitative as well as qualitative comparison of the proposed framework with recent methods.

### 4.1 DATASETS

In our experiments, we utilized the two challenging waste segmentation datasets namely ZeroWaste-f Bashkirova et al. (2022) and SpectralWaste Casao et al. (2024).

**ZeroWaste-f** Bashkirova et al. (2022) is a public waste segmentation dataset that poses significant challenges to the segmentation methods due to presence of translucent and deformable shaped waste objects in a highly cluttered environment. The dataset contains four types of waste objects that includes metal, cardboard, soft and rigid plastic. This dataset is divided into train, validation and test sets having 3002, 572 and 929 images respectively.

**SpectralWaste** Casao et al. (2024) is yet another challenging waste segmentation dataset that contains RGB and hyperspectral images. Similar to other works Ali et al. (2025), we utilized the RGB version of the dataset in our experiments for fair comparison. SpectralWaste is challenging in the sense as it contains translucent as well as thin elongated objects in cluttered scenes. The dataset contains six types of recyclable waste objects including basket, cardboard, film, filaments, video tape, and trash bags. SpectralWaste dataset is available in train, validation and test splits having 514, 167, and 171 images respectively.

### 4.2 EVALUATION METRICS

Similar to the existing works, we used *mean intersection over union (mIoU)* and *pixel accuracy* metrics to evaluate the performance of the proposed method on two waste segmentation datasets.

### 4.3 IMPLEMENTATION DETAILS

We implemented our method using MMSegmentation Contributors (2020) codebase and utilized Quadro RTX 6000 GPU for all experiments. We performed the experiments on a single GPU and used the batch

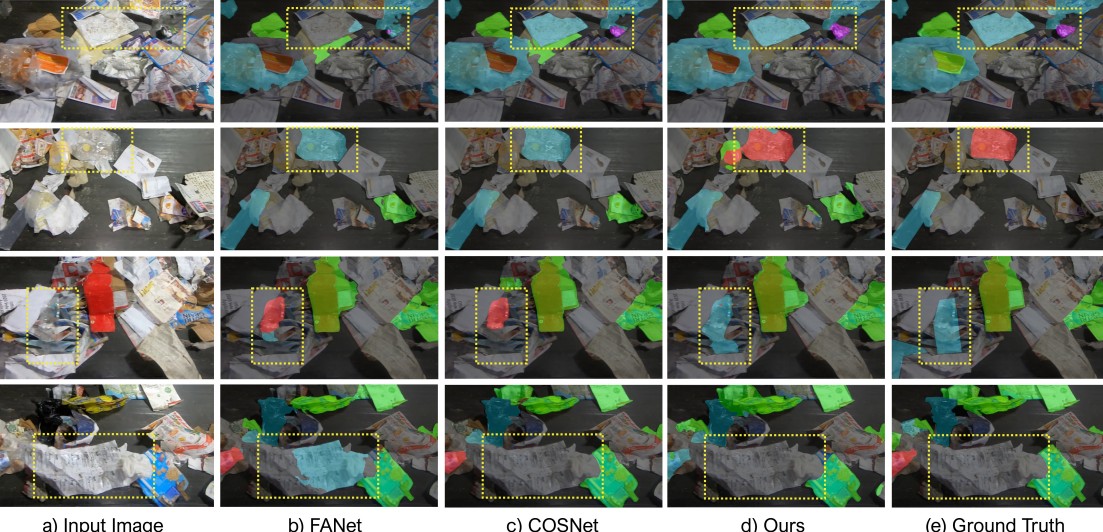

|  | a) Input Image | b) FANet | c) COSNet | d) Ours | (e) Ground Truth |

Figure 4: Qualitative comparison of EWSegNet with the recent waste segmentation methods FANet Ali et al. (2024) and COSNet Ali et al. (2025) on ZeroWaste-f. Proposed EWSegNet provides reasonably better segmentation performance as highlighted in yellow colored boxes.

Table 3: Performance comparison of EWSegNet with state-of-the-art waste segmentation methods in terms of mIoU (%) and class-wise IoU (%) scores on the test split of SpectralWaste Casao et al. (2024) dataset. Best results are highlighted in bold and second best are underlined.

| Method | mIoU (%) ↑ | IoU (%) ↑ | | | | | |
|---|---|---|---|---|---|---|---|
| | | Film | Basket | Cardboard | Video Tape | Filament | Trash Bag |
| MiniNet-v2 Alonso et al. (2020) | 44.5 | 63.1 | 58.9 | 55.4 | 30.6 | 10.0 | 49.2 |
| SegFormer-B0 Xie et al. (2021) | 48.4 | 66.9 | 71.3 | 48.9 | 33.6 | 15.2 | 54.6 |
| FANet Ali et al. (2024) | 67.83 | 72.47 | 82.98 | 75.26 | 41.28 | 67.65 | 67.36 |
| InternImage-T Wang et al. (2023) | 47.99 | 42.38 | 82.8 | 69.1 | 41.39 | 16.5 | 35.77 |
| COSNet | 69.96 | 77.61 | 83.65 | 75.14 | **42.95** | 69.06 | **71.38** |
| EWSegNet (Ours) | **70.59** | 77.86 | 84.16 | 77.94 | 41.77 | 72.88 | 68.95 |

size of 8. We initialize the encoder with pre-trained weights of ImageNet-1k Deng et al. (2009) and perform random initialization of decoder. Similar to the existing works Ali et al. (2024; 2025), we used Upernet Xiao et al. (2018) decoder in our model. During training, we utilized data augmentations of random resize, random crop to $512 \times 512$, and random horizontal flip. We utilized AdamW optimizer and an initial learning rate of 5e-5. We trained the model for 40k iterations on ZeroWaste-f and SpectralWaste datasets. During the evaluation phase, we resized the shorter side of the image to 512 as performed in training while keeping the aspect ratio.

### 4.4 QUANTITATIVE RESULTS

**Comparison on ZeroWaste-f:** We present the comparison of the proposed EWSegNet with the state-of-the-art (SOTA) waste segmentation methods Ali et al. (2025; 2024); Chen et al. (2018); Yang et al. (2022) on ZeroWaste-f dataset in Tab. 1. We observe that our method provides similar performance in terms of mIoU metric while significantly reducing the number of trainable parameters and the average latency (computed on test set of ZeroWaste-f dataset) as compared to the SOTA method COSNet. As demonstrated in Tab. 1, EWSegNet obtains mIoU score of 56.44% having number of encoder parameters equal to 23.3M and latency of 64.8 msec. Whereas the existing SOTA method COSNet provides the mIoU score of 56.67% while having encoder parameters equal to 27.3M and latency of 73.6 msec.

We further compare the class-wise IoU score of EWSegNet with the recent methods in Tab. 2. We notice that EWSegNet performs considerably better for the segmentation of metal class by achieving IoU gain of 5.44%. Overall, EWSegNet provides similar performance in terms of IoU while improving model efficiency.

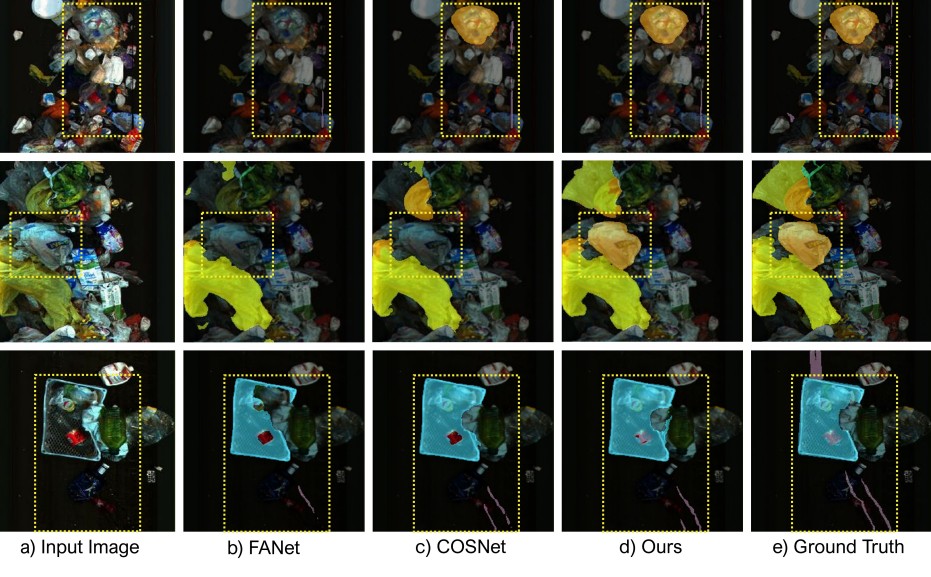

a) Input Image     b) FANet     c) COSNet     d) Ours     e) Ground Truth

Figure 5: Visual comparison of EWSegNet with recent waste segmentation methods FANet Ali et al. (2024) and COSNet Ali et al. (2025) on Spectral Waste dataset. As highlighted in yellow boxes, proposed EWSegNet is fairly better to segment the waste objects in cluttered scenes.

**Comparison on SpectralWaste:** In Tab. 3, performance comparison of the proposed EWSegNet and SOTA waste segmentation methods is reported. We observe that EWSegNet provides moderately better overall performance by achieving the mIoU score of 70.59% as compared to the SOTA method COSNet that obtains the mIoU score of 69.96%. It is worth notable that our method provides relatively better performance while being efficient.

Additionally, we observe that our method achieves relatively better IoU scores on four waste object types achieving an absolute gain of as high as 3.82% for *Filament* class. Whereas COSNet still performs better in case of *Video Tape* and *Trash Bag* classes.

### 4.5 QUALITATIVE RESULTS

**ZeroWaste-f:** Fig. 4 demonstrates the qualitative comparison of EWSegNet with SOTA waste segmentation methods including FANet Ali et al. (2024) and COSNet Ali et al. (2025). In the first row of the figure, FANet is unable to detect waste objects (overlaid with cyan and magenta) as highlighted in a yellow colored box, whereas our method and COSNet accurately segment the desired waste objects. In second and third rows, FANet and COSNet incorrectly classify the soft plastic objects as rigid plastic objects as shown in yellow boxes. We also notice that all the methods falsely segment the paper as cardboard object in second row of the Fig. 4 overlaid in green color.

**SpectralWaste:** The qualitative comparison of the proposed method with SOTA methods Ali et al. (2024; 2025) on SpectralWaste dataset is illustrated in Fig. 5. From the figure, we observe that the proposed EWSegNet and COSNet provide better performance compared to FANet by accurately segmenting the waste objects. As highlighted in the yellow box in first row of Fig. 5, FANet struggles to segment the waste objects whereas our method and COSNet provide fairly accurate segmentation of the waste objects such as video tape. Similarly, FANet struggles to accurately segment the desired waste objects in second and third rows while our method provides fairly better segmentation results as shown in yellow boxes.

### 4.6 ABLATION EXPERIMENTS

To verify the efficacy of the proposed modules, we performed ablation experiments by integrating the modules sequentially in the baseline. We report the performance of ablation experiments in Tab. 4. From the table, we observe that the baseline network (in which FCM, SCM and AFEM are missing and a large spa-

Figure 6: Here we demonstrate the efficacy of AFEM. a) shows the input image to EWSegNet, b) is the visualization of 3rd stage features which are input to AFEM, c) shows the visualization of the output features of AFEM, d) represents the boundaries highlighted by BE part in AFEM, e) shows the blobs emphasized by BA part in AFEM, f) is prediction of the EWSegNet, and g) shows the Ground Truth segmentation map.

Table 4: This ablation study reports the effectiveness of each proposed component of EWSegNet on ZeroWaste-f dataset. We observe that the baseline model provides reasonable performance. However, when FCM, SCM, and AFEM are integrated in the framework, the mIoU score is improved. Best results are highlighted in bold text.

|  | FCM | SCM | AFEM | mIoU (%)↑ | Pix. Acc. (%)↑ |
|---|---|---|---|---|---|
| Baseline | - | - | - | 47.32 | 90.77 |
| FCM | ✓ | - | - | 53.05 | 91.54 |
| FCM + SCM | ✓ | ✓ | - | 54.11 | 91.72 |
| EWSegNet | ✓ | ✓ | ✓ | **56.44** | **91.75** |

tial convolution is used) struggles to provide reasonable performance by achieving mIoU score of 47.32% on ZeroWaste-f dataset. When the baseline block is replaced with FCM, segmentation performance is increased to mIoU score of 53.1%. After integrating the spatial context module (SCM) with the FCM, segmentation performance improves by one percent. Finally, when the auxiliary feature enhancement module (AFEM) is utilized in EWSegNet, the mIoU score further improves to reach the value of 56.44% indicating the effectiveness of the proposed contributions.

**Efficacy of AFEM:** We further present the visual analysis of boundary enhancement and blob amplification in the auxiliary feature enhancement module in Fig. 6. As illustrated in the figure, features maps (see Fig. 6 b) which are input to the AFEM need to highlight the boundaries of the waste objects. AFEM takes these features and separately enhances the blobs regions and boundaries using BA and BE components resulting in emphasizing the desired semantic information as shown in Fig. 6 d) and e). Subsequently, the enhanced features from BE and BA are concatenated and fed to the convolution layer to obtain the highlighted waste objects as depicted in Fig. 6 c). This figure undoubtedly demonstrates the efficacy of the AFEM.

## 5   CONCLUSION

This work has introduced an efficient waste segmentation network (EWSegNet) that provides promising segmentation performance while reducing the computational cost of the network. The proposed EWSeg-Net processes the image in both spatial and frequency domains to capture the local and global contextual relationship between pixels without increasing the computational cost. In addition, the proposed EWSeg-Net uses AFEM to focus on the boundaries and desired semantic regions that are crucial for enhanced segmentation performance in cluttered scenarios. Furthermore, the mixing of the highlighted feature representation with third stage feature maps of encoder provides critical semantic information for the layers of last stage of encoder thereby improving the segmentation results. Extensive experiments on two challenging waste segmentation datasets demonstrate that EWSegNet can be utilized for automated waste recycling in complex scenes.

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
