# OpenReview forum: "A Step towards Efficient Waste Segmentation for Automated Waste Recycling in Cluttered Background"
_ICLR.cc/2026/Conference — ICLR 2026 Conference Withdrawn Submission_

### Official Review · Reviewer_uvPF · 2025-10-24

**Soundness:** 2
**Presentation:** 2
**Contribution:** 2
**Rating:** 4
**Confidence:** 4

**Summary:**

This paper proposes an efficient waste segmentation network, where the spatial context module enhances localized structural dependencies in the spatial domain, and the spectral context module subsequently captures global contextual relationships in the frequency domain. this cas caded design allows the network to progressively leverage both local and global representations across complementary domains to highlight the semantic information necessary for effective segmentation of waste objects in cluttered scenes. furthermore, the auxiliary feature enhancement focuses on structural information to enhance the target objects’ boundaries and blob amplification for better segmentation. experimentation on two challenging waste segmentation datasets including zerowaste-f and spectralwaste are conducted.

**Strengths:**

-  This proposes an end-to-end efficient waste segmentation network, called EWSegNet, which improves the model’s computational efficiency without compromising the segmentation performance.
- A frequency context module (FCM) that enables the model to learn data-dependent kernel and capture global context for better segmentation results is introduced.
- The spatial context module (SCM) performs the feature excitation and weighting to complement the FCM for rich feature extraction.
- The designed auxiliary feature enhancement module (AFEM) uses difference of Gaussian filtering and pooled attention to improve waste segmentation performance.
- Experimentation on two challenging waste segmentation datasets and demonstrate the effectiveness of the proposed method.

**Weaknesses:**

- I think the main weakness is limited novelty and marginal empirical gains. The core design—a spatial module followed by a frequency module plus a boundary/blob enhancement block—reads as an incremental combination of existing ideas (local spatial excitation, frequency-domain filtering, auxiliary sharpening), and the paper does not clearly articulate what is fundamentally new beyond engineering choices. Empirically the improvements over the closest competitor are very small.
- The Frequency Context Module is motivated by the frequency-domain equivalence between convolution and multiplication, but the paper does not provide indepth analysis or ablation that explains why the proposed multiplicative operation in Fourier space produces better semantic features.
- The authors emphasize efficiency and report encoder parameter counts and a latency number (EWSegNet: 23.3M params, 64.8 ms; COSNet: 27.3M, 73.6 ms). However, the improvement is also marginal.

**Questions:**

- Justification of the novelty.
- Justification of the marginal improvement.
- Justification of efficiency.

---

### Official Review · Reviewer_MMkU · 2025-10-31

**Soundness:** 3
**Presentation:** 1
**Contribution:** 3
**Rating:** 2
**Confidence:** 2

**Summary:**

This paper works on automated waste detection in a cluttered background. It proposes EWSegNet, which is an efficient waste segmentation network designed for automated recycling in cluttered environments. The method combines spatial and frequency domain processing to balance segmentation accuracy and computational efficiency. The network contains three key modules: spatial context module to capture the local dependencies using feature excitation and weighting; frequency context module to model global relationships in the frequency domain using fourier based operations; auxiliary feature enhancement module to improve boundary sharpness and blob segmentation using difference of Gaussian filtering and pooled attention. The authors have evaluated EWSegNet on two different datasets ZeroWaste-f and SpectralWaste. They have demonstrated that the segmentation accuracy of EWSegNet is comparable to prior state-of-the-art methods while reducing parameters and latency by about 15-20%. Ablation studies also support the contribution of each module.

**Strengths:**

The paper presents a well-engineered architecture that maintains comparable segmentation performance to prior methods while reducing model size and inference time, which is important to many real-world recycling systems. Moreover, the sequential use of spatial and frequency-domain processing is conceptually sound and well executed in the netword, this enables the model to capture both local and global contextual cues more efficiently.

**Weaknesses:**

The architecture builds on established ideas, such as spatial attention, frequency-domain filtering, and auxiliary enhancement. They are in an incremental way rather than introducing fundamentally new concepts.

The segmentation improvements over prior work, such as COSNet, are minor (only ~0.2% mloU). While the latency and number of parameters decrease and meaningful, the contribution feeld more engineering-oriented rather than conceptually innovative.

Several figures, for example, figures 2 and 3, contain vertically oriented labels and dense layouts that hinder readability. Citation format is very inconsistent (there are missing brackets for all citations that appear at the end of the sentence and redundant author names, such as Ali eta al, Ali eta al 2025). It should follow the ICLR standards.

The paper mainly compares to and talks about CNN-based methods as related work and baselines. However, recent works on waste classification using Vision Transformers and CLIP-based multimodal models explore similar application domains, such as (Xin Wang et al,  2023, Transformer-based automated segmentation of recycling materials for semantic understanding in construction). They should be mentioned in the related work section.

**Questions:**

Has the model been tested for robustness under variations such as lighting, transparency, or unseen waste materials? Would it generalize to datasets beyond the two tested in the paper?

---

### Official Review · Reviewer_RRS9 · 2025-11-01

**Soundness:** 2
**Presentation:** 2
**Contribution:** 1
**Rating:** 2
**Confidence:** 4

**Summary:**

This work focuses on the task of waste segmentation. Instead of using large vision foundation models, the authors propose a new architecture, EWSegNet, for efficient waste segmentation. Experiments are conducted on waste datasets.

**Strengths:**

Waste segmentation is useful for human life.

**Weaknesses:**

1. The authors highlight the efficiency of their network. But actually, no efficient metrics like GFLOPs are compared in the paper. No experiments can support their claims.

2. The designed modules are not novel to me. Lack of contributions in the methodology part. Specifically, why is spectral necessary to waste? No strong motivation for the designs of the modules in this work, which are not specific to the waste domain.

3.  If large vision models can achieve much higher performance, why not use them? With advanced hardware and technology, ViT can be efficient in real-world deployment.

4. DINOv3 pre-trained ViTs and other advanced networks should be compared. I believe these models would achieve much better performance than yours.

5. All the compared methods are old or on low-quality publications. Deeplab and focalnet are old. FANet and COSNet are published on ICIP and WACV, which are not top-tier conferences.

6. It seems that no waste segmentation works on top-tier publications are compared. It indicates this topic may not be sufficient for a top conference like ICLR.

**Questions:**

Please refer to the weakness.

---

### Official Review · Reviewer_giTi · 2025-11-01

**Soundness:** 2
**Presentation:** 2
**Contribution:** 2
**Rating:** 4
**Confidence:** 5

**Summary:**

The authors propose an efficient waste segmentation network, where the spatial context module enhances localized structural dependencies in the spatial domain, and the spectral context module subsequently captures global contextual relationships in the frequency domain.

**Strengths:**

- Authors propose a novel end-to-end efficient waste segmentation network, called EWSegNet, which improves the model’s computational efficiency without compromising the segmentation performance.
- Authors propose the frequency context module (FCM) that enables the model to learn data-dependent kernel and capture global context for better segmentation results.
- Authors introduce the spatial context module (SCM) that performs the feature excitation and weighting to complement the FCM for rich feature extraction.
- To emphasize the boundaries of the waste items and blob regions, authors design auxiliary feature enhancement module (AFEM) that uses the difference of Gaussian filtering and pooled attention thereby improving waste segmentation performance.
- Authors perform extensive experimentation on two challenging waste segmentation datasets and
demonstrate the effectiveness of the proposed method.

**Weaknesses:**

1) The article lacks a comparison with popular modern models that can solve the considered object segmentation problem, such as YOLOv11, YOLOv12, and YOLOE. They should have been taken into account and supplemented with them in the comparison tables.

[1] Khanam, R., & Hussain, M. (2024). Yolov11: An overview of the key architectural enhancements. arXiv preprint arXiv:2410.17725.

[2] Tian, ​​Y., Ye, Q., & Doermann, D. (2025). Yolov12: Attention-centric real-time object detectors. arXiv preprint arXiv:2502.12524.

[3] Wang, A., Liu, L., Chen, H., Lin, Z., Han, J., & Ding, G. (2025). Yoloe: Real-time seeing anything. arXiv preprint arXiv:2503.07465.

2) The article does not mention the well-known WARP dataset [1], which addressed the relevant problem of waste recognition.

[1] Yudin, D., Zakharenko, N., Smetanin, A., Filonov, R., Kichik, M., Kuznetsov, V., ... & Panov, A. (2024). Hierarchical waste detection with weakly supervised segmentation in images from recycling plants. Engineering Applications of Artificial Intelligence, 128, 107542.

3) The article's formatting should be improved; punctuation marks are missing at the end of the formulas.

4) No links to the anonymous repository could be found in the appendices or in the text of the article. This makes it difficult to verify the validity of the results for the model developed by the authors.

**Questions:**

1) What boundary cases can be demonstrated for the developed model? How are they explained? What are the limitations of the model's performance?

---

### Note · Authors · 2025-11-12

I have read and agree with the venue's withdrawal policy on behalf of myself and my co-authors.